# Introduction of Plant Transposon Annotation for Beginners

**DOI:** 10.3390/biology12121468

**Published:** 2023-11-26

**Authors:** Dongying Gao

**Affiliations:** Small Grains and Potato Germplasm Research Unit, USDA-ARS, Aberdeen, ID 83210, USA; dongying.gao@usda.gov

**Keywords:** transposon annotation, plant, genome, bioinformatics pipeline, database

## Abstract

**Simple Summary:**

Transposons are the most abundant repeats in plant genomes, and many of them can produce transcripts and encode proteins that may result in overestimating and incorrectly annotating functional genes. Thus, accurate transposon annotation is essential for all plant genome sequencing projects and other research. Although numerous tools have been developed, it is still challenging to annotate plant transposons for most scientists. The aims of this review are to introduce the basic knowledge about plant transposons and to provide a beginner’s guide on plant transposon annotation. I accentuate the importance of transposons and summarize the general strategies for transposon annotation. I briefly introduce the unique features of different transposon superfamilies in plants and the related resources for annotating plant transposons. I further present the information on improving the quality of transposon annotation. The challenges and future prospects for plant transposon annotation are also discussed.

**Abstract:**

Transposons are mobile DNA sequences that contribute large fractions of many plant genomes. They provide exclusive resources for tracking gene and genome evolution and for developing molecular tools for basic and applied research. Despite extensive efforts, it is still challenging to accurately annotate transposons, especially for beginners, as transposon prediction requires necessary expertise in both transposon biology and bioinformatics. Moreover, the complexity of plant genomes and the dynamic evolution of transposons also bring difficulties for genome-wide transposon discovery. This review summarizes the three major strategies for transposon detection including repeat-based, structure-based, and homology-based annotation, and introduces the transposon superfamilies identified in plants thus far, and some related bioinformatics resources for detecting plant transposons. Furthermore, it describes transposon classification and explains why the terms ‘autonomous’ and ‘non-autonomous’ cannot be used to classify the superfamilies of transposons. Lastly, this review also discusses how to identify misannotated transposons and improve the quality of the transposon database. This review provides helpful information about plant transposons and a beginner’s guide on annotating these repetitive sequences.

## 1. Introduction of Plant Transposons

Transposons or transposable elements (TEs) are genomic sequences that have the potential to change their positions in the host genome. According to their transposition mechanisms, transposons are grouped into two major classes: Class I elements or retrotransposons that move via a copy-and-paste model and Class II elements or DNA transposons that transpose via a cut-and-paste model, rolling-circle replication, or other mechanisms [1,2]. Each transposon class can be further divided into different superfamilies based on their sequence structures and the encoded proteins. Thus far, over 30 transposon superfamilies have been identified in prokaryotic and eukaryotic genomes [1,2,3,4,5,6]. However, only 16 superfamilies were found in the plant kingdom (Figure 1), including 2 superfamilies of long terminal repeat (LTR) retrotransposons, 3 superfamilies of long interspersed nuclear elements (LINEs), 1 superfamily of short interspersed nuclear elements (SINEs), and 9 DNA transposon superfamilies. Additionally, many plant genomes harbor endogenous plant pararetroviruses (EPRVs), which share similar core genes with LTR retroelements for reverse transcription but lack both functional integrase (INT) and LTRs [7,8]. Numerous elements can still be recognized as TEs as they show some typical features of transposons such as LTRs, terminal inverted repeats (TIRs), and flanking target site duplications (TSDs). However, they do not encode transposase (TPase) proteins and are difficult to group. These elements would usually be considered as unclassified transposons that include large retrotransposon derivatives (LARDs) [9], terminal-repeat retrotransposons in miniatures (TRIMs) [10,11], and miniature inverted-repeat transposable elements (MITEs) [12]. Notably, with the advance of sequencing technologies and related bioinformatics software, some novel types of transposons may be identified in the existing and/or newly sequenced plant genomes.

Transposons are important contributors for plant gene and genome evolution as their movements may alter gene expression and regulatory networks and result in chromosome rearrangements [13]. They can create novel genetic and morphological variations that are beneficial for the host’s fitness under disadvantageous environmental conditions. Transposons are key components of functional centromeres and play pivotal roles in the rapid divergence of centromeres between closely related plants [14]. TEs are frequently related to hybrid defects that might cause reproductive isolation across diverse species [15]. Transposons represent the most abundant repetitive sequences in plant genomes, and they are found in all sequenced plants including the model plant Arabidopsis and many major crops in the world such as rice (*Oryza sativa*), wheat (*Triticum aestivum*), barley (*Hordeum vulgare*), maize (*Zea mays*), and soybean (*Glycine max*) (Figure 2). Unlike some repeats in plant genomes, such as telomeric tandem repeats and ribosomal DNA (rDNA) repeats, most transposons are poorly conserved and can be quickly replaced in a relatively short period [16]. In many plants with larger genome sizes, transposons make up large fractions of the genomes. For example, about 85% of the maize genome is composed of transposons [17]. Therefore, transposon annotation is one of the most important and fundamental tasks for genome sequencing projects as it represents the precondition for many genomic analyses and the first step in the computation phase of genome annotation [18,19].

## 2. Strategies of Transposon Discovery

One of the most, or maybe the most, important goal of all plant genome sequencing projects is to accurately identify and annotate functional genes in the genomes. However, it is important to define and understand genes before the gene annotation is started. Generally, a gene is a transcribed unit of heredity that may or may not translate to a protein. It is well known that many TEs are expressed and can generate proteins. Thus, these transposon sequences can be treated as genes. However, except limited TEs called domesticated transposons, which have been co-opted by the host genome and confer some biological traits [20], the only function of the transposon-encoding proteins is for catalyzing transpositions. Therefore, TEs are frequently called ‘jumping genes’ or ‘selfish genes’, and they were excluded by many gene annotation projects which mostly focused on annotating the genes that encode non-transposase proteins and associate with biological and molecular functions and phenotypic traits, such as the genes controlling the yield, biotic, and abiotic tolerance, and other phenotypic traits in crops. 

Like genes and other genomic components, transposons are DNA sequences and are composed of four nucleotides including adenine (A), cytosine (C), guanine (G), and thymine (T). It is impossible to distinguish transposons from genes based on the composition of nucleotides. However, transposons are usually repetitive and exhibit unique structures; some of them may encode transposase proteins or reverse transcriptase (RT), which can be used for transposon prediction. Thus far, many resources including bioinformatics software and a TE database have been developed. Here, I just mention some of these TE-related software and databases as most of the resources were well documented in previous publications [21,22,23,24]. Generally, three major strategies are widely used for transposon annotations. 

### 2.1. Repeat-Based Annotation

Transposons are dispersed repeats, and many TE families are present in plant genomes in multiple copies; thus, these features can be used to develop bioinformatics tools for de novo transposon identification. One of the most popular programs is RECON, which identifies and groups TEs based on the pairwise alignments between genomic sequences [25]. To conduct genome-wide transposon annotations with RECON, the assembled genome sequences are usually split into smaller fragments, such as 10–20 Kb, and then used as the queries to search the whole genome sequences again to identify repetitive sequences and group them into different families. The represented element or consensus sequence of each repeat family are extracted, and the boundaries of the repeats can be defined via computational analysis and/or manual inspections (Figure 3A). For large plant genomes, such as wheat (~17 Gb), it is extremely challenging to conduct all-against-all comparisons. One practical option is to use a certain fraction (10–20% or more) of the genome to search against the whole genome sequences. However, this could miss some repeat families. It should be noted that not all transposon families are highly repetitive; some single-copy or low-copy transposons may not be detected by RECON. Additionally, many genes including gene families and duplicate genes are also scattered throughout plant genomes, and these repeats should be removed during transposon annotations. 

### 2.2. Structure-Based Annotation

Transposons exhibit some unique structural features such as LTRs, TIRs, and Poly(A) tails (Figure 1). Additionally, many transposon superfamilies can generate TSDs when they are moved and inserted into new genomic positions. Therefore, these unique characteristics can be used to develop bioinformatics programs for de novo transposon annotations. For example, many non-LTR retrotransposons contain a 3′ poly(A) terminal motif and are flanked by variable TSDs; thus, scientists can develop software to recognize the poly(A) motif, and the boundaries of non-LTR retroelements can be defined by TSDs. Then, the annotated sequences are extracted and grouped into different families based on their sequence comparisons (Figure 3B). Compared to RECON, structure-based software can annotate transposons with well-defined boundaries, and many of those programs are able to conduct genome-wide transposon annotations for the plants with larger genome sizes in a relatively short time. Thus far, most of the transposon annotation programs were developed based on the unique structures of distinct transposon superfamilies including those for annotating LTR retrotransposons [26,27,28,29,30,31], non-LTR retrotransposons [32,33], and DNA transposons [34,35,36,37,38,39,40,41]. Notably, the structure-based annotation software may find full-length transposons with both low and high copies, but they may miss some transposons without the typical transposon features. 

### 2.3. Homology-Based Annotation

This strategy is based on the hypothesis that transposons from related organisms could share common origins and show certain sequence similarities. Thus, researchers can use the known transposons which have already been annotated to find their homologous sequences in new genomes. The best-known and most widely used program for homology searches is RepeatMasker (https://www.repeatmasker.org, 1 November 2023), which applies customer or precompiled repeat libraries to identify homologous repeats. There are several ways to obtain known transposon sequences. The first is to check related transposon databases such as Repbase Update or Repbase, which is a well-curated and the most comprehensive database of repetitive elements in eukaryotic genomes [42]. Another way is to use the published transposon database from relative plants. For example, Garcia et al. (2021) used the annotated transposons in common bean (*Phaseolus vulgaris*) to find the homologous sequences in Lima bean (*Phaseolus lunatus* L.) [43]. The third way is to download deposited transposons from GenBank. However, the efficiency and accuracy of homology-based annotation heavily depend on the genomic similarity between the host plants of the reference transposons and the plants under study. As plant transposons are dynamic sequences and many TE families are species-specific, the reference transposons may yield poor annotations in distantly related plants. Also, this approach is not practical for the orphan plant lineages for which no transposon database is available in the relative species. Despite the fact that we can use the repeats in Repbase or other databases to find transposon-related sequences in distantly related organisms, in most cases, the homology sequences are short and fragmental and lack typical features of transposons. To obtain a better annotation, the proteins of identified transposons can be used to conduct a TBLASTN search against the newly sequenced plant genomes and to identify homologous sequences. Then, the hits and their flanking regions (5–10 Kb for each side) are extracted and used for BLASTN searches and genome-wide comparisons to identify complete transposons (Figure 3C). As a TBLASTN search is very sensitive in detecting homologous sequences, technically, it can identify more transposase-encoding elements including both full-length and truncated transposons as well as both single-copy and multiple copies of elements. 

Except the three major strategies above, other new methods of TE annotations were also developed, such as TASR, which is based on the fact that transposons are usually targeted and epigenetically silenced by 24 nt-siRNAs; thus, this feature can be used to develop a bioinformatics pipeline to recognize the regions targeted by the small RNAs and to define transposons [44]. Occasionally, some transposons inserted into genes and caused phenotypic variations. Thus, these transposons can be identified by cloning and comparing the gene sequences of mutants and the wild types. However, this method is not suitable for genome-wide transposon discovery. As plant genomes are very complicated and each of the transposon annotation methods has its own limitations, it is better to combine multiple computational pipelines to obtain high-quality transposon annotations. 

## 3. Steps for Transposon Annotation

There are three basic steps in genome-wide transposon annotation: preliminary annotation, classification, and quality check and data improvement. However, many bioinformatics pipelines for transposon annotation combine the first and second steps and can directly output classified transposons. 

### 3.1. Brief Introduction of Plant Transposon Superfamilies and Their Annotations

#### 3.1.1. LTR Retrotransposons

LTR retrotransposons represent the most abundant repeats in many, maybe most, plant genomes, such as maize, which consists of about 75% LTR retroelements [17]. These retroelements exhibit some structural hallmarks including LTR, TSD, reverse transposase proteins, primer binding site (PBS), polypurine tract (PPT), and the 5′TG…CA3′ terminal motifs that can be used for characterizing LTR retrotransposons. Many plant transposon annotations usually start with LTR retrotransposons for three reasons: (1) LTR retrotransposons contribute large fractions of plant genomes; (2) several excellent bioinformatics programs have been developed [26,27,28,29,30,31] for annotating LTR retrotransposons; and (3) this software may generate lower error rates of data mining than many programs for the de novo detection of DNA transposons and non-LTR retrotransposons. By following the protocols of the developed software, users can generate sufficient data for preliminary predictions of LTR retrotransposons. In plant genomes, only two superfamilies of LTR retrotransposons, *Ty1/Copia* and *Ty3/Gypsy*, are found; these two superfamilies share similar structures but have a different order of the reverse transcriptase and integrase (INT) domains (Figure 1). Most complete LTR retrotransposons are large (about 4–10 Kb) and some can be over 20 Kb. However, some types of LTR retroelements called TRIMs are small (about 250 bp to 1500 bp) and have tiny LTRs (<100 bp) [11]. Therefore, users need to pay attention to the default parameters of the related software. For LTR_Finder, the default minimum size for LTRs and the internal regions is 100 bp and 1000 bp, respectively [27]. To obtain better annotations with LTR_ Finder, we usually use the sets ./ltr_finder -d 30 -D 15000 -l 30 -L 5000 -s ./tRNAdb/Athal-tRNAs.fa where d represents the minimum distance between 5′ and 3′ of the LTRs, D represents the maximum distance between 5′ and 3′ LTRs, l represents the minimum size of 5′ and 3′ LTRs, L represents the maximum size of 5′ and 3′ LTRs, and s means the tRNA sequence file we used. 

#### 3.1.2. LINEs 

LINEs are non-LTR retrotransposons and may be the most ancient types of retrotransposons in plant genomes. The typical LINEs usually contain non-LTR reverse transposases and the poly(A) terminal motif. The general methods for LINEs annotations are presented in Figure 3, and the related software including RECON [25] and MGEScan-Non-LTR [45] can be used for automatic LINE annotations. The boundaries of full-length LINEs can be defined by TSDs. However, most LINE elements in plant genomes are truncated at their 5′ end due to unsuccessful reverse transcription or other reasons. It should be noted that RTE retrotransposons, which is one superfamily of LINEs, lack a poly (A) tail and instead contain tandem repeats [46,47]. It seemed that the RTEs in flowering plants were horizontally transferred from aphids or other unsequenced animals [47]. As the RTEs from different plants show a high sequence similarity, thus, the identified RTEs can be used to find the homologous elements in new plant genomes. 

#### 3.1.3. SINEs 

SINEs are another type of non-LTR retrotransposon. Unlike LINEs, they are very small (<500 bp) and lack conserved coding domains. Also, the 5′ end of many SINEs are truncated. Thus, SINEs may represent the superfamily that is most difficult to precisely annotate among all plant transposons. Nearly all plant SINEs were derived from tRNAs; they usually contain an internal RNA polymerase III (pol III) promoter consisting of box A and box B motifs, and the 3′ poly (A) flanking by TSDs [48]. Thus far, some SINE annotation software has been developed to recognize these structural motifs and conduct genome-wide SINE prediction [32,33]. SINEs were found in a wide range of plants, many of which are present in specific lineages, such as p-SINE1 in the *Oryza* genus [49], but some SINE families such as Au are widely distributed in both dicots and monocots [50] that provide good resources for homology-based SINE annotations [51]. 

#### 3.1.4. Endogenous Plant Pararetrovirus (EPRVs)

Pararetroviruses are double-stranded DNA viruses that replicate through an RNA intermediate. They were named by Temin (1985) to distinguish hepadnaviruses in animals from retroviruses that are RNA viruses and integrate their DNA copies into host genomes [52]. Except very few members, such as *Petunia vein-clearing virus* (PVCV) [53], the vast majority of plant pararetroviruses do not have the integrase domain. All plant pararetroviruses belong to the family *Caulimoviridae*, they contain one to eight open reading frames (ORFs), and their genome sizes range from 7.1 to 9.8 Kb [54]. Endogenous pararetroviruses (EPRVs) are present in the genomes of a wide range of plants [55,56], and they encode polyproteins that share a high sequence similarity with Ty3/Gypsy LTR retrotransposons; thus, EPRVs were frequently misannotated as Ty3/Gypsy LTR retrotransposons or were ignored. All EPRVs identified thus far lack LTRs [7,8], which brings difficulties in annotating these pararetroviral sequences and accurately defining their boundaries. Recently, the bioinformatics pipeline CAULIFINDER has been developed for the automatic annotation and classification of EPRVs in plant genomes [7].

#### 3.1.5. DNA Transposons with TIRs 

Among the nine superfamilies of DNA transposons identified in plant genomes, seven superfamilies have TIRs including Mutator, CACTA, hAT, PIF/Harbinger, Tc1/Mariner, Sola, and Ginger. 

##### Mutator Transposons

Mutator transposons or mutator-like transposable elements (MULEs) were originally discovered in maize [57] and have since been identified in other plants, animals, and fungi [58]. The transposons of this superfamily usually carry relatively long TIRs (can be over 300 bp) and produce TSDs of 7–12 bp [59]. MULEs are abundant in plant genomes and may represent the most mutagenic plant transposons identified thus far [60]. They are usually less than 5 Kb in size and contain one ORF-encoding mutator transposase. However, MULEs can be over 8 Kb and may have multiple ORFs including one encoding transposase protein and other(s), which may help transposon movement, or their functions are not very clear [60,61]. Some MULEs called Pack-MULEs have captured functional genes or fragments of expressed genes and play important roles in plant gene evolution [59,62]. 

##### CACTA Transposons

CACTA transposons were first found in maize and named *Enhancer* (*En*) and *Suppressor-mutator* (*Spm*) [63,64]. The typical features of this superfamily are the terminal motifs starting with CACTA or CACTG and ending with TAGTG or CAGTG. The TIRs of CACTA elements are short (mostly < 50 bp) and they generate 3-bp TSDs. The complete CACTA transposons are usually large (>10 Kb), and some of them can be over 20 kb in size [65]. Due to their large sizes and high copy numbers, CACTA transposons contribute larger fractions of many plant genomes than other DNA transposons. Like Pack-MULEs, some CACTA transposons have been found to carry host gene sequences [66].

##### hAT Transposons

hAT transposons were first identified by Barbara McClintock as the Activator or Ac element [67], but the name of this superfamily is an acronym of its three members: hobo from Drosophila, Ac from maize, and Tam3 from snapdragon [68]. hAT elements are widely distributed in plants and other eukaryotes, they are typically less than 5 kb in length, and usually contain short TIRs (5–27 bp) flanked by 8-bp TSDs [69,70]. In addition, the transposase of many hAT elements contains the conserved domain of 50 amino acids at the C terminus, which may be involved in dimerization or other functions [71]. 

##### PIF/Harbinger Transposons

The PIF/Harbinger transposon superfamily obtains its name from the two founding members: the P instability factor (PIF) from maize [72] and Harbinger from *Arabidopsis thaliana* [73]. They usually contain short TIRs (about 14–50 bp) and generate 3-bp TSDs (TAA or TTA). Unlike other DNA transposons, the intact PIF/Harbinger elements have two main open reading frames (ORFs); one ORF produces catalytic transposase containing a conserved DDE motif and another ORF encodes a Myb-like protein [74]. PIF/Harbinger transposons have been found in the genomes of many plants and some of them were co-opted or domesticated to serve as new molecular functions associated with yield and other traits in plants [75,76]. 

##### Tc1/Mariner Transposons

Tc1 and Mariner transposons was first discovered in Caenorhabditis elegans and Drosophila mauritiana in the 1980s, respectively [77,78], and then were found in a wide range of organisms including both prokaryotes and eukaryotes [58]. The Tc1/Mariner transposons in plants mostly range in size from 1.5 to 6 Kb and contain relatively short TIRs (~30 bp) [79,80]. In contrast to other TEs that create variable TSDs, the Tc1/Mariner superfamily always generates 2-bp TSDs of TA. They are not very abundant in many plant genomes and were missed by some plant genome annotations. 

##### Sola Transposons 

Sola transposons were first reported in 2009 and they are distributed in a wide range of organisms including bacteria and metazoans [3]. The transposons of the Sola superfamily encode DDD-TPase and are flanked by 4-bp TSDs. Most Sola elements range in size from 2 Kb to 9 Kb, but some can be over 15 Kb. They also have very variable terminal sequences and the TIRs of the reported Sola transposons ranged from 11 bp to 1124 bp [3]. Despite Sola transposons being found in some plants, such as the moss *Physcomitrella patens*, it seems that they are not common in flowering plants. 

##### Ginger Transposons

Gypsy INteGrasE Related (Ginger) transposons are unusual DNA transposons as they contain TIRs and a protein which shares a high sequence similarity to the integrase encoded by Gypsy LTR retrotransposons [4]. DNA transposons containing a Gypsy-like integrase were first found in *Dictyostelium discoideum* [81,82], and subsequently identified in animals [4] and plants [5]. The TIRs of Ginger transposons are relatively long (40–270 bp) and contain the 5′-TG…CA-3′ terminal dinucleotides, and the insertions of Ginger transposons generate TSDs of 4–6 bp [4,5]. 

##### MITEs 

MITEs were first identified in maize in 1992 [12], and they are small DNA transposons (mostly < 500 bp) with TIRs surrounded by variable TSDs. In contrast to many previously reported DNA transposons that were mobile with the ‘cut and paste’ model and were usually present in low copy numbers, MITEs were frequently found in high copy numbers, which can be over 10,000 copies for some families [83]. Additionally, dramatic copy number differences for the same MITE family can be detected between closely related genomes [74], suggesting that MITEs can rapidly increase their copy numbers in a short period. Therefore, MITEs were considered as a special type of DNA transposons or even Class III transposons [84]. However, more studies revealed that MITEs were likely derived from the internal deletions of large DNA elements as the TIRs of MITEs showed a high sequence identity to that of several reported DNA transposon superfamilies such as PIF/Harbinger and hAT [74,85,86,87]. 

##### TIR DNA Transposon Annotation

Several programs have been developed to annotate TIR DNA transposons such as TIRvish [34], TIR-Learner [35], Generic Repeat Finder [36], and Inverted Repeat Finder [88]. The software provides good resources to detect transposons solely based on the recognition of TIRs or the combination of structure and homology transposon annotations. In addition, MITE-Hunter [37], detectMITE [38], MiteFinderII [39], MITE Tracker [40], and other related bioinformatics pipelines can be used to find small TIR-DNA transposons that lack transposases. As the TIRs of DNA transposons are usually short and less conserved, it is still challenging to accurately detect these DNA transposons and define their boundaries. 

#### 3.1.6. Helitron Transposons

Helitron transposons were first identified in Arabidopsis, rice, and nematode (*Caenorhabditis elegans*) in 2001 [89]. They have conservative termini (5′-TC … CTRR (mostly CTAG)-3′) and contain hairpins (16–20 bp) separated by 10–12 nucleotides from the 3′ end. However, all Helitron elements do not have terminal inverted repeats and do not generate TSDs; they transpose precisely between 5′AT3′. Helitron-like sequences are widely present in plant genomes; some of them are large and can capture gene fragments [90]. Helitron transposons are difficult to annotate and precisely define their boundaries as they lack both TSDs and TIRs. They can be detected using the unique terminal motifs and the helicase sequences or with related software such as HelitronScanner [41].

#### 3.1.7. Replitron Transposons

Replitron is a new group of DNA transposon reported in 2023 [6]. They lack TIRs but contain short direct terminal repeats ranging from 5 to 11 bp and HUH endonuclease that is distantly related to Helitron transposons. Replitron elements are present in the genomes of green algae, liverworts, mosses, lycophytes, and ferns, but absent in hornworts and seed plants [6]. The identified Replitrons are 900 bp to 4 kb, and their insertions can generate TSDs of 2–8 bp. Thus far, no software has been developed for annotating Replitron transposons. One practical approach is to conduct homology searches with the endonuclease proteins of known Replitrons and identify new Replitron sequences based on sequence comparisons. 

#### 3.1.8. Others TE Annotation Resources

Despite the programs used to predict specific types of transposons that have been developed, there is no need to annotate each of the transposon superfamilies one by one. Several bioinformatics packages have been created through combining multiple programs and methods to annotate all types of transposons, such as Extensive de-novo TE Annotator (EDTA) [91], RepeatExplorer2 [92], TransposonUltimate [93], Earl Grey [94], and other comprehensive bioinformatics pipelines. 

### 3.2. Classification of Transposons

Once the potential transposon sequences have been identified, the next step is to group them and define their families. Many bioinformatics pipelines can annotate and automatically generate classified transposons. Here, I just describe how to handle the transposon sequences that are not grouped. 

#### 3.2.1. Definition of Transposon Superfamilies

The superfamilies of transposons can be defined based on the encoded transposase proteins and other sequence features including TIRs, LTRs, and the sizes of TSDs. To classify the transposon superfamilies, the widely used method is to use the generated transposon sequences for conducting a BLASTX or BLASTN search against GenBank or other databases such as Gypsy Database (GyDB) [95] that provide a valuable resource to define the superfamilies of retroelements and determine the conserved domains of LTR retrotransposons and EPRVs. However, many annotated transposons have accumulated mutations or undergone deletions and encode no transposase or short protein sequences, and the superfamilies for these TEs can be defined based on their terminal motifs and flanked TSDs, or they can be classified as unclassified transposons, such as TRIMs, LARDs, or MITEs. It is important to note that unclassified transposons should be real transposons and non-transposon sequences should be excluded. 

#### 3.2.2. Definition of Transposon Families

In some cases, it is challenging to define transposon families as different scientists applied distinct cutoffs or strategies. Thus far, transposon families are commonly defined using the ‘80-80-80’ rule, which means that any members of the same transposon family should be over 80 bp and show more than an 80% sequence identity over 80% of their sizes [2]. However, other studies argued that this definition may ignore the consensus sequences [96] and may not be applicable for monophyletic groups [97]. As many transposons may contain highly conserved regions, such as the internal RT domain of LTR retrotransposons, transposons of the same family defined by the ‘80-80-80’ rule may be grouped into different phylogenetic clades. The terminal sequences including both LTRs and TIRs are less conserved than the transposase-encoded regions, and elements of the same transposon family usually exhibit extensive sequence identity at their termini. Therefore, the terminal sequences of transposons may be more sensitive in defining transposon families. In addition, members from the same transposon family should share close phylogenetic relationships and should be catalyzed by the same transposase enzymes. The better strategies for defining a transposon family should consider sequence identity and phylogenetic origins [98] as well as the molecular interactions. 

#### 3.2.3. Autonomous and Non-Autonomous Transposons 

Transposons can also be grouped into autonomous and non-autonomous elements. The former are usually complete transposons and encode the entire enzymes required for transposition, whereas the latter may have undergone internal deletions or mutations and lack functional transposases and their movement is catalyzed by their autonomous partners. It should be noted that the terms ‘autonomous’ and ‘non-autonomous’ are defined based on the functional mobility, and they cannot be used to classify superfamilies of transposons. Any superfamily of both Class I and II transposons can be divided into either autonomous or non-autonomous elements. In many cases, autonomous and non-autonomous transposons were defined solely based on in silico gene prediction; if any transposons encode transposase proteins and contain the entire domains for transposition, they can be generally considered as autonomous transposons. However, the accuracy and reliability of computational predictions depend on many factors. If the mobility of a transposon has not been experimentally validated, any autonomous elements defined by computational prediction should be treated as potentially (but not true) autonomous transposons even their gene models are well supported by transcriptional data. 

### 3.3. Quality Control and Improvement of Transposon Annotation

Transposons are highly repetitive, and many plants have complex genome structures, especially those with large genome sizes and/or duplicated chromosomes. Despite many bioinformatics software or pipelines having been developed and used for transposon prediction, not all annotated sequences are real transposons and some of them may be misannotated. Additionally, some protein-coding genes can also be predicted as repetitive sequences by some de novo annotation tools [19]. Therefore, users must carefully evaluate the annotated sequences, discard the incorrect annotation, and improve the quality of transposon annotation when they obtain the first version of the transposon database for newly sequenced plant genomes. Computational analysis is important, but manual inspection is strongly encouraged for this step. Goubert et al. have proposed a guideline for the manual curation of transposons [99]. Here, I just want to highlight four things that are helpful for improving the quality of transposon annotation. 

#### 3.3.1. Identification and Exclusion of Misannotated Sequences

Generally, three major types of genomic sequences were frequently misannotated: tandem repeats (TRs), non-transposon sequences, and misclassified transposons. Occasionally, some transposons such as TRIMs were tandemly organized [11]. However, nearly all tandem repeats are long track repeats such as centromeric tandem repeats, they are widely dispersed, and consist of various basic units. These sequences were frequently misannotated as LTR retrotransposons and significantly reduce the accuracy of transposon annotations. TRs can be identified with related programs [100] or pairwise sequence alignments (Figure 4). Non-transposon sequences, such as genes, pseudogenes, and other genomic sequences, can also be misidentified as transposons, especially for those annotated as fragmental or unclassified transposons. Two methods can be used to identify these types of misannotations: one is to use these annotated sequences for BLASTX or BLASTN searches and see if they show a significant sequence similarity to the known transposon protein sequences; and another is to extract the annotated sequences and the flanking regions and check if they are repetitive and contain any sequence structures of transposons (TIRs, LTRs, and TSDs). In some cases, some transposon sequences were misclassified. For example, I manually inspected a SINE database annotated by a computational program and found that some annotated SINEs were fragments of true LTR retrotransposons. 

#### 3.3.2. Elimination of Nested Transposons

One transposon called bottom transposon may contain other transposon(s) named nested transposon(s). Nested organizations of transposons are very common in many plants with higher repeat contents such as maize [101,102]. They are also frequently found in some specific regions of several plant genomes with lower repeat contents such as the centromeric and pericentromeric regions in rice [14]. In some cases, transposons were organized into multiple layers of nested insertions in which nested elements further served as target transposons for other TEs [14,102]. In plants, LTR retrotransposons frequently acted as target transposons to harbor other LTR retrotransposons of the same or different families, but DNA transposons can also be inserted by LTR retrotransposons and other transposons. Despite nested transposons providing some insights into the evolutionary dynamics of transposons, they may cause problems for transposon classifications. Therefore, nested transposons should be removed from the reference transposons. Nested transposons can be easily identified by comparing different family members of the same transposon family or orthologous elements between multiple genomes from the same or related species. Despite identification of inserted transposons being tedious and time-consuming, this process may find some novel transposon families and many of them may be recently active. 

#### 3.3.3. Definition of Transposon Boundaries

The exact boundaries of transposons can be determined based on the terminal sequences, including LTR and TIRs, and TSDs. However, the movements of some transposons such as Helitrons do not generate TSDs. Additionally, not all transposons possess the typical features of their superfamilies, some of them may have accumulated mutations or undergone deletions, and their termini and TSDs are difficult to recognize. Unfortunately, no software is available for defining the boundaries for fragmental or mutated transposons. One strategy is to extract the flanking sequences of transposons and conduct sequence alignment analysis. It is exhausting and time-consuming but is practical. With the advance of sequencing technologies, multiple genomes from the same and closely related species have been sequenced, and the exact boundaries of many transposons can also be defined with multiple reference sequence sets [103]. 

#### 3.3.4. Identification of Unannotated Transposons 

Due to the complexity of plant genomes and the current limitations of available bioinformatics resources for de novo TE discovery, some transposons may be missed. There are several ways to identify missing transposons. The first method is to apply more and different annotation approaches and see if any new transposons can be discovered. The second strategy is to check the reported transposons in newly sequenced genomes. Transposons are usually very dynamic, but some transposons are present in a wide range of plants. For example, An-RTEs (RTE clade of LINEs), Cassandra (TRIMs), and Au-SINEs are present in many dicots and monocots [11,47,50,104]. If they are not found in a new sequenced dicot or monocot, one possibility is that they may be missed by the transposon annotations. The third way is to use the annotated transposons to screen the genome with RepeatMasker and use the transposase proteins of identified superfamilies to conduct a TBLASTN search against the masked genome. If multiple significant hits were detected, it suggested that some transposons must be missed. 

### 3.4. Criteria for Good Transposon Database

One question is how to evaluate the quality of the annotated transposon database? If two research groups (A and B) independently annotated transposons in the same genome, the A group’s transposons can define 75% of the genome whereas the B group’s transposons only mask 70% of the genome. Can we say the transposon database of the A group was better than that of the B group? The answer for this may be yes or no. The coverage is an indicator, but not the sole one to evaluate the quality of transposon annotation. In my opinion, there are four criteria to judge a good transposon database. a. Transposons should be accurately annotated. Two major points for this are (1) the annotated transposons should be real transposons and non-TE sequences should be excluded; (2) all annotated transposons should be correctly classified. For example, a Ty3/Gypsy retrotransposon cannot be classified as a Ty1/Copia. In some publications, the majority of the annotated transposons were listed as unknown/unclassified/ungrouped transposons, which implied the extremely poor quality of transposon annotations and/or genome sequencing and assembling. b. The boundaries of transposons should be well defined. c. Nearly all transposons should be annotated. It is impossible to identify all transposons in plant genomes, but we should not miss many of them. d. The transposon database should be non-redundant. Transposons are very redundant, and it is not necessary to include all or many copies of a transposon family into the database as it will enlarge the size of the TE library and impede computational analyses; thus, it is better to have a well-defined reference transposon for each family. 

## 4. Conclusions and Future Directions

Despite extensive research having been conducted, the accurate annotation of transposons remains challenging and time-consuming. Thus far, 16 superfamilies of transposons have been found in plants and numerous bioinformatics software and pipelines have been developed to annotate transposons based on the three major annotation strategies. Although different systems or parameters were proposed for transposon classification and family definition, the terms ‘autonomous’ and ‘non-autonomous’ cannot be used to classify transposon superfamilies. A good transposon database should be accurately annotated and fully represented, and the boundaries of transposons should be well defined. As transposon annotation heavily depends on the available genomes, it is impossible to have a high-quality TE database with poorly sequenced and assembled genomes. With the innovation in software development and sequencing techniques, more complete plant genomes will be well annotated, and new types of transposons may also be discovered in plants. The current bioinformatics programs offer incredible resources for transposon predictions; however, some of them are difficult to install and use for beginners, especially for those with little expertise in bioinformatics and computational biology, and many of these programs also produce numerous false-positive annotations. New methodologies and algorithms are needed to develop next-generation annotation tools for significantly improving the accuracy and efficiency of transposon annotation. It is highly appreciated for beginners to create user-friendly transposon annotation platforms via national and international collaborations and to allow users to submit their genomes and download the annotated transposons online. Another need is to develop bioinformatics tools for comparatively annotating transposons with multiple genomes and to identify more new and unshared transposons. Thus far, multiple genomes from the same species and/or related species have been sequenced for many important crops or model plants; however, the comprehensive annotation and comparison of transposon landscapes in the sequenced pan-genomes are very limited. It is also highly recommended to periodically update the current transposon databases and make them to be accessible publicly. 

## Figures and Tables

**Figure 1 biology-12-01468-f001:**
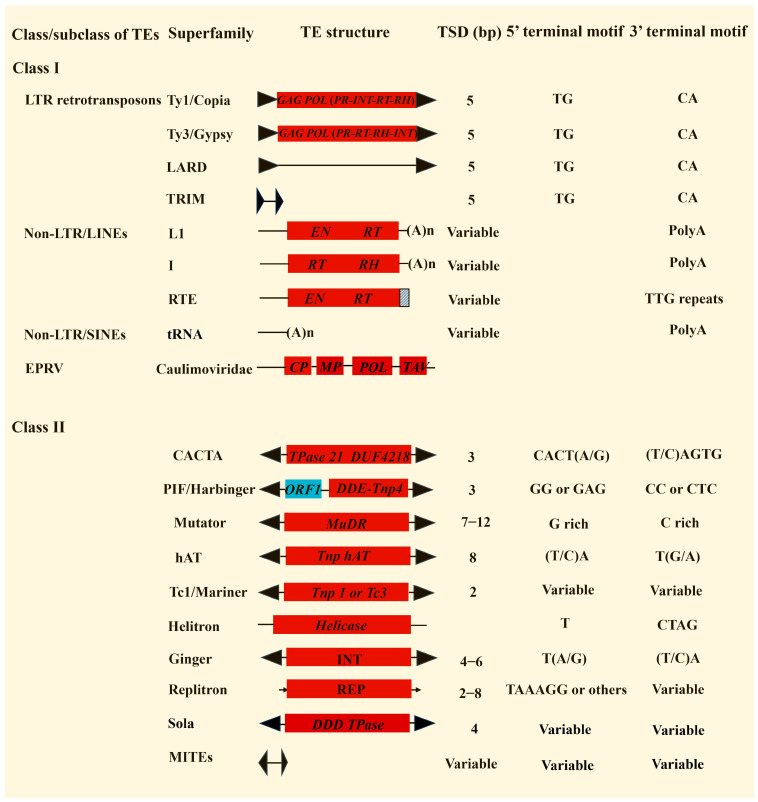
Structures and typical features of different types of plant transposons. GAG, group-specific antigen; POL, polyprotein; PR, protease; RT, reverse transcriptase; RH, RNAase-H; INT, integrase; EN, endonuclease; CP, coat protein; MP, movement protein; TAV, transactivation protein; TPase, transposase; REP, replication protein.

**Figure 2 biology-12-01468-f002:**
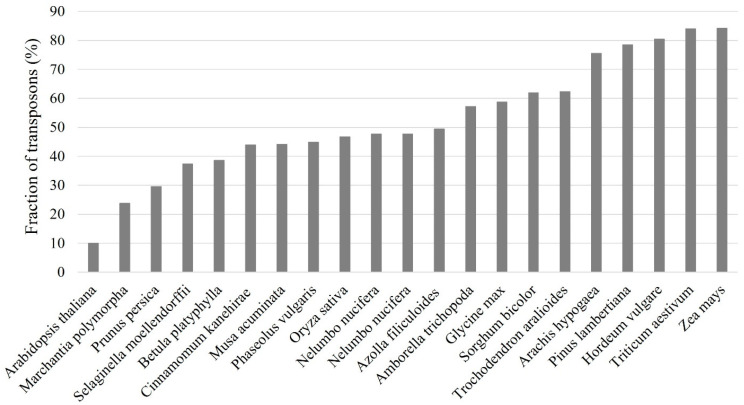
Transposon fractions of 21 sequenced plant genomes.

**Figure 3 biology-12-01468-f003:**
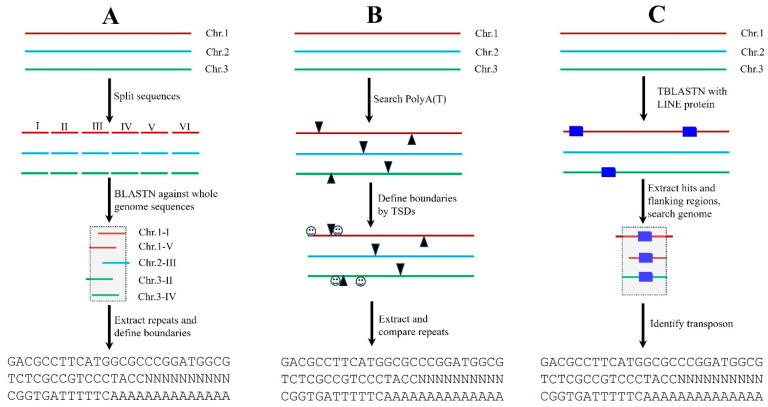
Three major strategies for annotating transposons (LINEs as the example). (**A**) Repeat-based transposon annotation. The plant genome sequences (assembled chromosomes or large scaffolds) were split into smaller fragments, and then used to search against all plant genome sequences. All repetitive sequences were extracted and grouped into different repeat families based on sequence identity. The boundaries of transposons can be defined by computational comparisons. (**B**) Structure-based transposon annotation. The poly(A) motif of LINEs was recognized by developed software and the boundaries of LINEs were defined based on TSDs (smiley faces). Then, the identified LINE sequences were extracted and grouped into different families. (**C**) Homology-based transposon annotation. The proteins of represented LINEs were used to search against the plant genomes and to identify homologous sequences (blue rectangles). Then, the homologous hits and the flanking regions (usually 5–10 Kb for each direction) were extracted together and used to conduct sequence comparisons for defining boundaries of LINEs and grouping them into families.

**Figure 4 biology-12-01468-f004:**
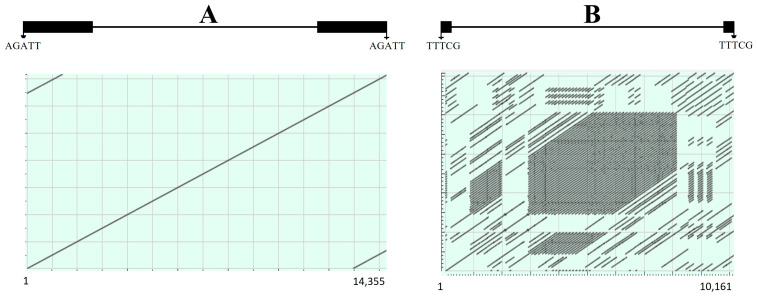
Pairwise sequence comparisons of two potential LTR retrotransposons in barley annotated by bioinformatics software. Both annotated sequences have LTRs (black rectangles) and were flanked by 5-bp TSDs. However, pairwise comparisons revealed that sequence A was a real LTR retroelement whereas sequence B was misannotated as it contained tandem repeats.

## Data Availability

Not applicable.

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
