# Peer review of "Introduction of Plant Transposon Annotation for Beginners"

_biology, 2023, doi:10.3390/biology12121468_

Round 1
Reviewer 1 Report
Comments and Suggestions for Authors
Repeatexplorer site and tools (Novak, P., Neumann, P., Macas, J. (2020) – Global analysis of repetitive DNA from unassembled sequence reads using RepeatExplorer2. Nature Protocols 15:3745–3776.)
the tools run on galaxy thus, upon simple registration and login, can be run very intensive tasks and so may appear especially useful for beginners devoid of appropriate hardware. Many family level annotation pipelines implement, e.g. ,DANTE repeatexplorer tool which can be especially helpful for classification at family level.
also, as family level classification can be especially hard and it would be important to define alternatives and ambiguities e.g as in Neumann et al. Mobile DNA (2019) 10:1 https://doi.org/10.1186/s13100-018-0144-1
Comments on the Quality of English Language
line 67: substitute "close related plants" with "closely related plants"
line 333: substitute "contract" with "contrast"
line 382 : seems more appropriate substitue " generated" with "identified"
Author Response
Many thanks for your valuable comments, please see my responses below:
1). Repeatexplorer site and tools (Novak, P., Neumann, P., Macas, J. (2020) – Global analysis of repetitive DNA from unassembled sequence reads using RepeatExplorer2. Nature Protocols 15:3745–3776.) the tools run on galaxy thus, upon simple registration and login, can be run very intensive tasks and so may appear especially useful for beginners devoid of appropriate hardware. Many family level annotation pipelines implement, e.g. ,DANTE repeatexplorer tool which can be especially helpful for classification at family level.also, as family level classification can be especially hard and it would be important to define alternatives and ambiguities e.g as in Neumann et al. Mobile DNA (2019) 10:1 https://doi.org/10.1186/s13100-018-0144-1
Response: Added the two references as Reference 92 and 97.
2). line 67: substitute "close related plants" with "closely related plants"
Response: Corrected.
3). line 333: substitute "contract" with "contrast"
Response: Corrected.
4). line 382 : seems more appropriate substitue " generated" with "identified"
Response: Changed.
Reviewer 2 Report
Comments and Suggestions for Authors
The manuscript titled “Introduction of Plant Transposon Annotation for Beginners” summarizes what the transposable elements are, their classification, the different approaches traditionally used for their identification, as well as the implications of their correct annotation on a genome, particularly with a focus on plant genomes. The manuscript provides references to bioinformatic tools dedicated to the annotation of specific transposable elements, which will be useful for users who not only need to mask repetitive sequences on a genome but also need to characterize a specific class of transposable elements. In general, the manuscript is well written, although I consider some minor changes that could be made to improve its readability.
- In Figure 1, the text inside of red boxes is difficult to read, a different color background for the boxes will improve this figure.
- In Figure 2, the scientific names need to be italicized
- In lines 122 through 124, it would be helpful for the readers if there were some examples of gene or gene families, that should be removed during the annotation of TEs.
- The paragraph starting from lines 212 to 216 can be improved. Especially at the beginning of the paragraph, where my understanding is that the author means that there are two superfamilies of LTR in plants (copia and Gypsy).
- In line 233, it may be helpful to describe that RTE is one class of LINEs since they were not mentioned before in the manuscript.
- The sentence from lines 247 to 249 can be improved by adding some commas. For instance, “families, such as Au, “
- In line 333, “In contract” perhaps means “In contrast”.
- The paragraph from lines 333 to 336 can be improved by adding some commas. For instance, “model, which usually …”
- In line 374, I consider this subsection to need to be renamed, since the word others can be interpreted as different classes of TE that were not described in the previous sections. Especially since this section refers to bioinformatic tools that can annotate multiple classes of TEs.
- Section 3.4 is valuable. The author desires to clarify that there are several factors to consider to evaluate the quality of a TE database. However, it needs to be rewritten to be more concise and solve a few grammar issues. For instance, line 509, “One question is that how to evaluate the quality of transposon database?” could be re-written as: “One question is, how to evaluate the quality of the transposon database?” or how to evaluate the quality of the transposon database?
Comments on the Quality of English Language
I found some minor grammatical issues, which are mentioned in my general comments.
Author Response
Thanks a lot for your helpful comments, please see my responses below:
1). In Figure 1, the text inside of red boxes is difficult to read, a different color background for the boxes will improve this figure.
Response: Thanks for the good point, we changed the colors for both boxes and background, hopefully, the figure can be clearly read.
2). In Figure 2, the scientific names need to be italicized
Response: Changed.
3). In lines 122 through 124, it would be helpful for the readers if there were some examples of gene or gene families, that should be removed during the annotation of TEs.
Response: I met this situation when I was annotating the transposons in common bean during 2012-2013. However, I didn’t keep the raw data for such long time.
4). The paragraph starting from lines 212 to 216 can be improved. Especially at the beginning of the paragraph, where my understanding is that the author means that there are two superfamilies of LTR in plants (copia and Gypsy).
Response: I modified the description “. In plant genomes, only two superfamilies of LTR retrotransposons, Ty1/Copia and Ty3/Gypsy, are found, these two superfamilies share …”
5). In line 233, it may be helpful to describe that RTE is one class of LINEs since they were not mentioned before in the manuscript.
Response: I added “which is one superfamily of LINEs”.
6).The sentence from lines 247 to 249 can be improved by adding some commas. For instance, “families, such as Au, “
Response: I added commas “, such as Au,”
7). In line 333, “In contract” perhaps means “In contrast”.
Response: Changed.
8). The paragraph from lines 333 to 336 can be improved by adding some commas. For instance, “model, which usually …”
Response: I removed “usually does not increase new copies” and modified the description “that mobile with the ‘cut and paste’ model and are usually present in low copy numbers”.
9). In line 374, I consider this subsection to need to be renamed, since the word others can be interpreted as different classes of TE that were not described in the previous sections. Especially since this section refers to bioinformatic tools that can annotate multiple classes of TEs.
Response: I modified the title to “Other TE annotation resources” which including some bioinformatics packages that can annotate all TE superfamilies at same time.
10). Section 3.4 is valuable. The author desires to clarify that there are several factors to consider to evaluate the quality of a TE database. However, it needs to be rewritten to be more concise and solve a few grammar issues. For instance, line 509, “One question is that how to evaluate the quality of transposon database?” could be re-written as: “One question is, how to evaluate the quality of the transposon database?” or how to evaluate the quality of the transposon database?
Response: I modified the description to “One question is, how to evaluate the quality of the annotated transposon database?”
Reviewer 3 Report
Comments and Suggestions for Authors
The study of transposons is one of the hot topics in the field of genomics. Transposons are the main factors of genome variation and play an important role in gene expression regulation, biological trait formation and species differentiation. However, due to the complexity of the genome, it is still a great challenge for beginners to accurately identify the transposon components and their content in the genome. Therefore, this review is necessary and helpful for beginners. However, as a review, the logic and structure of this article are flawed.
1. For the review of transposons, the author should introduce TE and its importance, classification, and identification difficulties, and then introduce identification strategies and representative software. However, the full text only mentions a few software or algorithms, so this article is up to the standard of knowledge, but the introduction of methods is relatively short.
2. The author needs to use a table to list most of the mainstream software involved in each identification strategy, and summarize the advantages and disadvantages of each software, applicable scenarios, etc., so that it is more convenient for beginners to choose software according to their own needs.
3. TE is a research hotspot, and a lot of software and algorithms have been developed for its identification. Therefore, there will be a lot of references in this review article. However, the number of references in this article is relatively small, which makes it difficult to fully cover the current research progress, and the content of this article needs to be enriched.
4. Line 402: Consider providing a brief explanation or context for the '80-80-80' rule when it is introduced. This will help readers understand its significance in defining transposon families.
5. In section 3.3, provide more context on how misannotated sequences are identified. Additionally, consider elaborating on the implications of misannotations in transposon databases.
6. Section 4: Summarize the key points of the manuscript in the conclusion section, emphasizing the significance of accurate transposon annotation and the challenges that remain. The section on future directions should be expanded to provide more insights or suggestions for potential improvements or advancements in transposon annotation methods.
Author Response
Thanks for your critical but helpful comments, please see my responses below:
- For the review of transposons, the author should introduce TE and its importance, classification, and identification difficulties, and then introduce identification strategies and representative software. However, the full text only mentions a few software or algorithms, so this article is up to the standard of knowledge, but the introduction of methods is relatively short.
Response: I provide the related information on “TE and its importance, classification, and identification difficulties, and the identification strategies and representative software” in the MS. However, as the Title of the MS indicates that this is an introduction for beginners (But not a protocol). The goal of this review is to generally introduce transposons and transposon annotations. Also, as many bioinformatics programs need to install and run multiple software, it will be very hard for readers/beginners to understand if I listed the specific commands.
- The author needs to use a table to list most of the mainstream software involved in each identification strategy, and summarize the advantages and disadvantages of each software, applicable scenarios, etc., so that it is more convenient for beginners to choose software according to their own needs.
Response: The tables listed related transposon annotation software have been summarized in previous publications (see the references 21-24). I noted these in the MS “Thus far, many resources including bioinformatics software and TE database have been developed. Here, I just mention some of these TE-related software and databases as most of the resources were well documented in the previous publications [21-24].”
Regarding the point for listing the advantages/disadvantages of related software, it is a good point, but it is not practical. As many bioinformatics software/programs/packages have been developed, no scientists can say they are familiar with all these pipelines. It is hard to make a conclusion on the advantages and disadvantages of each software if scientists never used. In this review, I have listed some software/programs/websites include some I are using and recommending, see the section “3.1. Brief introduction of plant transposon superfamilies and their annotations”
- TE is a research hotspot, and a lot of software and algorithms have been developed for its identification. Therefore, there will be a lot of references in this review article. However, the number of references in this article is relatively small, which makes it difficult to fully cover the current research progress, and the content of this article needs to be enriched.
Response: I feel that I have listed the related references in the MS.
- Line 402: Consider providing a brief explanation or context for the '80-80-80' rule when it is introduced. This will help readers understand its significance in defining transposon families.
Response: I provided the information in the MS, see “using the ‘80-80-80’ rule which means that any members of a same transposon family should be over 80 bp and show more than 80% sequence identity over 80% of their sizes [2]”.
- In section 3.3, provide more context on how misannotated sequences are identified. Additionally, consider elaborating on the implications of misannotations in transposon databases.
Response: I added some words “Two methods can be used to identify these types of mis-annotations, one is to use these annotated sequences for BLASTX or BLASTN searches and see if they show significant sequence similarity to the known transposon protein sequences; and another is to extract the annotated sequences and the flanking regions and check if they are repetitive and contain any sequence structures of transposons (TIRs, LTRs and TSDs).”
- Section 4: Summarize the key points of the manuscript in the conclusion section, emphasizing the significance of accurate transposon annotation and the challenges that remain. The section on future directions should be expanded to provide more insights or suggestions for potential improvements or advancements in transposon annotation methods.
Response: Thanks for the good point. I added more words “It is highly appreciated for beginners to create user-friendly transposon annotation platforms by national and international collaborations and to allow users to submit their genomes and download the annotated transposons online. Another need is to develop bioinformatics tools for comparatively annotating transposons with multiple genomes and to identify more new and unshared transposons. Thus far, multiple genomes from the same species and/or related species have been sequenced for many important crops, however, comprehensive annotation and comparison of the transposon landscapes in the sequenced pan-genomes are very limited.”.
Round 2
Reviewer 3 Report
Comments and Suggestions for Authors
The author responded to my reviews with more justification than revision. After all, writing a paper is not easy. Well, you can publish this article, and I have no further comment.